# Methodical Considerations and Resistance Evaluation against *Fusarium graminearum* and *F. culmorum* Head Blight in Wheat. Part 3. Susceptibility Window and Resistance Expression

**DOI:** 10.3390/microorganisms8050627

**Published:** 2020-04-25

**Authors:** Andrea György, Beata Tóth, Monika Varga, Akos Mesterhazy

**Affiliations:** Cereal Research Non-Profit Ltd., 6701 Szeged, Hungary

**Keywords:** resistance expression, aggressiveness, *F. graminearum*, *F. culmorum*, isolate effect, disease index, *Fusarium*-damaged kernel, deoxynivalenol, susceptibility window, inoculation time and FHB response

## Abstract

Flowering is the most favorable host stage for *Fusarium* infection in wheat, which is called the susceptibility window (SW). It is not known how long it takes, how it changes in different resistance classes, nor how stable is the plant reaction in the SW. We have no information, how the traits disease index (DI), *Fusarium*-damaged kernel rate (FDK), and deoxynivalenol (DON) respond within the 16 days period. Seven winter wheat genotypes differing in resistance were tested (2013–2014). Four *Fusarium* isolates were used for inoculation at mid-anthesis, and 4, 8, 11, 13, and 16 days thereafter. The DI was not suitable to determine the length of the SW. In the *Fusarium*-damaged kernels (FDK), a sharp 50% decrease was found after the 8th day. The largest reduction (above 60%) was recorded for DON at each resistance level between the 8th and 11th day. This trait showed the SW most precisely. The SW is reasonably stable in the first 8–9 days. This fits for all resistance classes. The use of four isolates significantly improved the reliability and credit of the testing. The stable eight-day long SW helps to reduce the number of inoculations. The most important trait to determine the SW is the DON reaction and not the visual symptoms.

## 1. Introduction

*Fusarium* head blight (FHB) is one of the most destructive diseases of wheat (*Triticum aestivum* L.) worldwide. Research in the last decades clarifies that the most important toxin-regulating agent is disease resistance [1,2,3,4]. Therefore, most of the work belongs to the competence of plant breeding. The artificial inoculations have a larger significance as the natural conditions do not support enough selection work, and this is true also for research. It is long known [5,6] that an increased susceptibility exists during the flowering period, called the susceptibility window (SW). In essence, this is what we know now about this important feature. A possible explanation is that the pollen contains compounds (betaine, choline) stimulating germination of *Fusarium* spores, thus inducing more aggressive disease development [7].

Investigations in inoculation timing have been done in both greenhouse and field experiments. Schroeder and Christensen [8] inoculated in a greenhouse seven spring wheat cultivars at anthesis, milk, and soft dough stages, with spray and single-spikelet inoculation. They concluded that the degree of resistance to both initial infection (type I resistance) and the spread within the spike after infection (type II resistance) could be determined when the inoculation time is known, i.e., artificial inoculation is needed for both. Schroeder and Christensen [8] recorded disease symptoms through evaluation the rate of diseased spikelets and percentages following spray and point inoculation; their data correspond to the disease index. However, they did not find significant differences between resistances to them; the difference for type II resistance (point inoculation) was much lower than *Fusarium*-damaged kernel rate FDK values that were found. Therefore, from our context it is important to include FDK to better understand the role of FDK in the resistance process. Hart et al. [9] spray-inoculated the plants of cultivar Genesee in a greenhouse and seven winter wheat cultivars in a field experiment with a single *F. culmorum* isolate. In the greenhouse experiment, the inoculations were performed at the following developmental stages: <¼ filled, ¼–½ filled, ½–¾ filled, and fully filled. The later inoculations were beyond the flowering time. The early stages were not extensively studied. Inoculation timing in the field ranged from early watery to mid-dough stage. Spikes were covered with plastic bags for varying periods of time following inoculation in both experiments. Deoxynivalenol (DON) production occurred in wheat spikes under favorable moisture conditions, even at late stages of kernel development, while yield reductions were greatest when infections occurred before kernel filling. In a one-year study under a controlled environment, Lacey et al. [10] inoculated one winter wheat cultivar at different time points from spike emergence to harvest using several species of *Fusarium* with varying mist duration after inoculation. Anthesis was the only infection time for which high DON concentrations were observed and both disease severity and DON content sharply decreased for inoculations performed after mid-anthesis. However, significant interactions occurred between infection timing and moisture duration following inoculation, such that DON levels from mid-anthesis infections rose more sharply with increasing postinoculation moisture durations. Del Ponte et al. [11] inoculated a susceptible wheat genotype using a single *F. graminearum* isolate at six stages from mid-anthesis to hard dough in a greenhouse experiment. The percentage of damaged kernels was >94% for inoculations performed between mid-anthesis and late milk stages, but fell to 23% for inoculation at hard dough. The highest DON concentrations were found in samples inoculated at the watery-ripe and early milk stages, but DON was still detected at later stages. In field experiments, Cowger and Arrellano [12] inoculated eight winter wheat cultivars with four *F. graminearum* isolates at mid-anthesis by spray inoculation. The inoculum was a mix of equal proportions of spores of the four isolates. Additional inoculations were made at 10 and 20 days after mid-anthesis in four experimental years. In three of the four years, infections at 10 days after anthesis produced less FDK than infections at anthesis. DON levels were as high from infections at 10 days after anthesis as from infections at anthesis in two years, but lower in two other years. Siou et al. [13] sprayed the highly susceptible winter wheat cultivar Royssac with eight isolates of *F. graminearum*, *F. culmorum* and *F. poae* in a greenhouse experiment. Even the isolate x inoculation time interaction was significant, the authors did not make any comment on this finding. Four dates of inoculation were tested: anther extrusion, 8 days post-anther extrusion, 18 post-anther extrusion (milky kernel development stage) and 28 post-anther extrusion (dough development stage). The highest disease and toxin levels were for inoculations around anthesis, but late infections led to detectable levels of fungus and toxin for the most aggressive isolates. Mesterhazy and Bartók [14] compared fungicide efficacy at full flowering and 10 days thereafter by spraying inoculation. Surprisingly, between disease severity and toxin contamination, no significant difference was found. However, at the first inoculation, the weather was cooler and then later with more rain; there is only weak proof that the window is ten days long. Therefore, the ten days could have only been a random factor.

More isolates in a test are used when aggressiveness of the isolates should be investigated. In serial tests for resistance and phenotyping, only one isolate [15] or a mixture of isolates [10,16,17] are used e.g. one aggressiveness level was applied. Occasionally, the spawn method was also used [18]. In some cases, no source of inoculum is specified [19]. As *F. graminearum* and *F. culmorum* do not have specialized races [20,21] as does, for example, *Puccinia striiformis*, the general conviction is that all inocula are equally good for testing. Two points were missed: (1) there is a variability within species for aggressiveness (the aggressiveness level strongly influences the differentiation of the genotypes) [3,4,22,23], and (2) the aggressiveness of the isolates is not stable, which was proven by tests where more independent isolates were used on different genotypes [2,3,4,23,24]. As in earlier tests, we worked with four independent isolates, and the isolate role was not highlighted; Therefore in this paper we present details in this matter.

In summary, the papers reviewed confirmed that anthesis is the most susceptible stage for *Fusarium* infection. It also became clear that even late infections can result in high harvest-time DON levels, but the papers do not provide solid information on the length of the SW and its stability within this period, as only two inoculation times represented the early period. For us, it was a problem as to whether or not we have a stability reaction. When we inoculate according the optimal time, we need to inoculate on every second or third day. As weather is seldom stable, every inoculation time might give large differences; the comparison of the data may result in artifacts. This jeopardizes the presentation of the comparable data that makes problems everywhere, in breeding, in genetic analysis, variety registration, etc. When the SW would be stable for a week, it is enough to inoculate once a week and a much higher number of genotypes could be inoculated that will have the same ecological conditions. As we do not have well-supported facts, and it is not known whether the different traits respond similarly or differently, there remained an unsolved problem. We have seen that visual symptoms, FDK, and DON contamination often differ [3,4,22,24,25] Therefore, their role in resistance estimation needed further illumination. The use of one or more inocula was not clarified in the past decades, so we need further research in understanding their role. Therefore, inoculations were planned by using four isolates (two *F. graminearum* and two *F. culmorum*) and seven winter wheat genotypes to provide a deeper understanding of the SW and utilize results in resistance testing at six inoculation dates, from mid-anthesis to 16 days thereafter.

Objectives of this study were: (1) to determine the length of SW in field conditions; (2) to determine the role of visual head symptoms (DI), *Fusarium*-damaged kernels (FDK), and the DON content in forming the length of SW; (3) to describe the role of resistance level in the SW response; and (4) to determine how the results can be used to improve the reliability of the resistance testing.

## 2. Materials and Methods

### 2.1. Plant Material

Seven genotypes were selected with differing resistance to FHB, including four winter wheat cultivars from the variety breeding program at Szeged without strong selection for FHB, Hungary, and three lines chosen from our FHB resistance program (Table 1). They had similar flowering times, but differed in FHB resistance level. According to the previous tests, two genotypes proved to be susceptible, one was rated as moderately susceptible, two were rated as moderately resistant, and two were rated as resistant.

### 2.2. Field Experiments for Disease Evaluation

The plant material was evaluated in the nursery of the Cereal Research Nonprofit Ltd. in Szeged, Hungary, (46°14′24″ N, 20°5′39″ E) over two seasons (2012/2013 and 2013/2014). The field experiments were conducted in three replications in a randomized complete block design. A plot (genotype, replicate) consisted of 12 rows, 1.5 m long with a 20 cm row spacing. For the six inoculation times, two row subplots were used. For control, the first inoculation served as the mid-anthesis inoculation (Figure 1). The seven cultivars and three replicates produced 21 plots for an experiment. Each genotype per replications was sown in six (one plot per inoculation date) two-row plots of 1.5 m length in mid-October, using a Wintersteiger Plot Spider planter (Wintersteiger GmbH, Ried, Austria). The width of the plots was approximately 40 cm.

### 2.3. Inoculum Production and Inoculation Procedure

Fungal suspensions were made in 10 L heat-stable glass flasks filled with 9.4 L liquid Czapek–Dox medium [26]. They were aerated at room temperature for a week. Suspensions were then stored at 4 °C until they were used for inoculation. Inocula contained mycelium and conidia as well. Each genotype was inoculated individually with two *F. culmorum* (F.c. 12375 /1/, F.c. 52.10 /2/) and two *F. graminearum* (F.g. 19.42 /3/, Fg. 13.38 /4/) isolates. The same isolates were used for studies in 2013 and 2014. The cited authors used a different conidium concentration/mL. Schroeder and Christensen [8] applied an inoculum with 5–8 × 10^6^ conidia/mL, Hart et al. [9] used 6.6 × 10^4^ conidia /mL, Cowger and Arrellano [12] used 1 × 10^5^ conidia/mL, and Siou et al. used 2 × 10^4^ conidia/mL [13]. This suggests that there is no agreement for which concentration is optimal. The authors did not report the reason for using the given concentration. Theoretically, it should have been some relation with aggressiveness, but none intended to give an explanation. This means that chances were low to find a good solution. However, an aggressiveness test [26] was developed to measure directly the aggressiveness of the given inoculum. This solved the problem and we could test this very important feature. A test needed 10 Petri dishes, 5–5 for two genotypes in seedling stage. Sterile double layer filter paper was placed in 12 cm diameter dishes, 9 mL suspension was uniformly spread on the surface and 25 seeds were placed into the dish in a 5 × 5 binding. Besides the original concentration, 1:1, 1:2, and 1:4 dilutions were also applied, along with a control with sterile distilled water. The rating was made daily from the second to sixth day. The number of healthy germs was evaluated. From the test, the aggressiveness could be directly seen. Therefore, the selection of the inocula was made in this system. As we did not want to relate the aggressiveness of the isolates used to each other, the standardized conidium concentration was not necessary. Even so, the four isolates differed significantly from each other.

Six inoculation dates were tested, the first at anthesis at Feekes growth stage 10.5.1 [27], second at four days, third at eight days, fourth at 11 days, fifth at 13 days, and the sixth 16 days later. Inoculation started on about 10 May. The inoculation was performed with the spray inoculation method [23]. Bunches of 15–25 spikes were sprayed from all sides with a handheld sprayer with the use of 15–20 mL fungal suspension for each sample, as described by Mesterhazy [23,28].

After inoculation, bunches were covered for 48 h in a transparent polyethylene bag, sealed to maintain 100% relative humidity and promote infection. After removing the bags, the plants were loosely bound with a label for identification at half-plant height to allow the leaves to photosynthesize freely.

### 2.4. Disease Assessment

Evaluation of FHB disease severity started on the 10th day after inoculation and was repeated on 14th, 18th, 22nd, and 26th day, until the control heads started to become yellowish. The percentage of infected spikelets was rated for the whole group of heads. On average, 24 spikelets were in a head. When one spikelet was infected in each head, this was 4%, when only one was in the bunch of 15, then this was 0.3% [28], all percent data mean disease index (DI). This was sensitive enough to provide a comparable data series to FDK and DON data. Their arithmetical mean values across all observations per groups of heads were used as entries for statistical analysis. At maturity (beginning in July), the groups of inoculated heads were harvested manually and stored in paper bags. The samples were threshed using a stationary thresher (Wintersteiger LD 180, Ried, Austria) at low wind, in order to retain the shriveled *Fusarium*-damaged kernels (Mesterházy, 1987, 1995). Chaff was separated using an Ets Plaut-Aubry (41290 Conan-Oucques, France) air separator. FDK was rated visually (estimation of scabby grains with definite tombstone, or chalky-white and rose discoloration as a percentage). All of the work was finished at the end of July.

### 2.5. Deoxynivalenol Analysis

Of the grains in a group of heads, six grams were separated and milled by a Perten Laboratory Mill (Type: 3310, Perten Instruments, 126 53 Hagersten, Sweden). From this, 1 gram was used for toxin extraction for each replicate of each inoculation date in the case of each isolate with 4 mL of acetonitrile/water (84/16, *v*/*v*) for 2.5 h with a vertical shaker. Following centrifugation (10,000 rpm, 10 min), 2.5 mL of the extract was passed through an activated charcoal/neutral alumina SPE column at a flow rate of 1 mL/min. Thereafter, 1.5 mL of the clear extract was transferred to a vial and evaporated to dryness at 40 °C under vacuum. The residue was dissolved in 500 µL of acetonitrile/water (20/80, *v*/*v*). HPLC separation and quantification were performed on an Agilent 1260 HPLC system (Agilent Technologies, Santa Clara, California, USA), which was equipped with a membrane degasser, a binary pump, a standard autosampler, a thermostatted column compartment, and a diode array detector (DAD). DON was separated on a Zorbax SB-Aq (4.6 × 50 × 3.5 µm) column (Agilent) equipped with a Zorbax SB-Aq guard column (4.6 × 12.5 × 5 µm) thermostatted at 40 °C. The mobile phase A was water, while mobile phase B was acetonitrile. The gradient elution was performed as follows: 0 min, 5% B; 5 min, 15% B; 8 min, 15% B; 10 min, 5% B; 12 min, 5% B. The flow rate was set to 1 mL/min. The injection volume was 5 µL. DON was monitored at 219 nm. This procedure updated the methodology used by Mesterhazy et al. (1999) [3]. The only difference was that the measurements were made by a new Agilent Infinity 1260 HPLC system.

### 2.6. Statistical Analysis

Correlation analysis was performed using the functions provided by Microsoft Excel 2013’s Analysis ToolPak add-in program. The statistical evaluations for four-way ANOVA followed the functions by Sváb (1981) [29] and Weber (1967) [30], with the help of the Microsoft Excel background. The significance between the main effect and the interactions were developed by functions from Weber [30], the *df* values in ANOVA for these are given in Table 2.

## 3. Results

The visual data that are generally considered as resistance data did not show any decrease in the 16-day inoculation period (Table 3). At the 8th and 16th day, larger values were found, but values found on the other four days were at the same level. This means that the visual scores did not show any sign of the susceptibility window in the first 16 days, i.e., the whole period acted as a long susceptibility window without signs of decrease, even following the flowering; a decrease was anticipated. The resistance differences between genotypes were highly significant; F569/Kö was the best at 5.18% and GK Futár was the most susceptible at 18.9% disease index. The cultivars reacted rather inconsistently, the most resistant varieties gave the same numbers across the whole period, but others showed variable performance, as shown by the diverging correlation coefficients (Table 3A). On the other hand, the correlations between variety reactions and different inoculation dates were much higher, and only several were not significant at *p* = 0.05. This would indicate a reasonable stability of responses across the different inoculation dates.

The FDK data (Table 4A) showed a different picture. Between mid-flowering and eight days after inoculation, except the fourth day (it is significantly lower than the initial and eighth day values), stability was mostly seen, thereafter a sharp, more than 50% reduction was recorded, followed by an additional one-third decrease that remained stable afterwards also by the 16th day. This decrease is highly significant, with limit of significant difference (LSD) 5% at 1.76%. The cultivar differences between means were five-fold, the most resistant showed 5.43% FDK and the most susceptible showed 25.1%. The variation width was near 20%, the LSD 5% value was 1.91, so a roughly ten-fold difference existed between this and the variation width. This meant a rather stable reaction in the first eight days, experimental proof to the presence of the susceptibility window. We should say that the correlations between genotype reactions (Table 4B) were much closer than they were in the FHB disease index, all correlations were significant between *p* = 0.05 and *p* = 0.001. This indicated a similar response of the cultivar reactions in the 16 day inoculation period. The reduction can be seen at all resistance levels. The rate was similar, but the data were much lower in the more resistant genotypes, which explains the closer correlation values. The correlations between the response of the genotypes to different inoculation data were also similar (Table 4C), but in several cases the level was lower, and three correlations were not significant.

The DON data (Table 5A) for the 0–8th day showed a stable mean DON contamination between 9.23 and 9.81 mg/kg concentration. Most of the cultivars were stable, and the differences between inoculation dates were not large. The decrease of DON contamination was very sharp after the eighth day, only one-third remained from the eighth day amount. This decreased by an additional 50% to the 13th day and a further significant decrease was recorded between the 11th and 13th day. The additional small decrease up to the 16th day was not significant.

Therefore, counting by FDK and FDK, the susceptibility window lasted for the same eight days for all resistance classes. Also, different cultivar reactions can be recognized. The most resistant F569/Kő had the lowest DON at about 3 mg/kg, and this could decrease the DON level below the EU limit of 1.25 mg/kg. The case of the GK Fény variety is important, since in 2010 the large FHB epidemic damaged it only moderately in contrast to GK Garaboly, which had the highest DON contamination at flowering. We observed a similar phenomenon with GK Csillag; its resistance was better than GK Fény. Attention should be paid to the fact that in normal testing only the first inoculation was made and the others were not existing. The data show also that to see the resistance behavior for delicate and highly important cultivars, the 11th- and 16th-day inoculation is suggested. By this way we could detect cultivars like GK Csillag, that was rather susceptible at mid-anthesis and could be discarded based on disease index, but in later inoculation dates it gave the second place following the most resistant genotype. Correlations between genotype reactions were very close, as it was for FDK, but at a higher level (Table 5B). Therefore, cultivars across inoculation times behaved similarly. We should also consider that good correlations existed for the DON data only between the first four inoculation dates, during the later inoculation periods no significant correlations were found (Table 5C). This means that the resistance relationships do not automatically follow this range later on, so therefore unexpected phenomena may occur. Only one example, GK Csillag, had 10 ppm DON content at mid-flowering and 0.64 ppm on the last inoculation. F569/Kő started with 2.98 and finished with 0.57 mg/kg, practically the same value. The DON data in graphical form (Figure 2) shows the genotype reaction during the six inoculation periods. The DON regulation seemed to be more complicated than usually expected.

The general means of the isolate data showed rather large differences for DI, but for all isolates the genotype differences were significant (Figure 3). Fg 19.42 isolate was the most aggressive; the others displayed only 40–50% of its value. The FDK data showed larger isolate differences than DI, the resistance differentiation less expressed for the three less aggressive isolate (Figure 4). We had the largest difference between isolates for DON compared to DI and FDK (Figure 5); the Fg 19.42 isolate kept its excellent performance, the Fc 12375 showed yet significant differences, and the two least aggressive isolates did not show a proper differentiation, as they showed for DI and FDK. It seems that even at medium aggressiveness in DI or FDK can cause a problem in DON response. 

Looking at the three traits across cultivars, we observed a clear picture of how the FHB data did not show any sign of decreasing during the 16-day period. However, the FDK and DON showed much similarity (Figure 6). The correlation for disease index/FDK was r = −0.1639, DI/DON was r = −0.2347, and FDK/DON was highly significant at r = 0.9673 with *p* = 0.01. This is again an argument that FDK is a more important trait to characterize resistance level than DI.

The general means for the genotypes and traits showed several tendencies (Figure 7). The most resistant F569/Kő showed the lowest values for all three traits. The F569/81.F.379 had an FDK value twice as large compared to the FHB than the most resistant genotype. As we have the mean values of six inoculations, the resistance data are more reliable than only with one inoculation at full flowering stage. Thus, this two-fold difference does not seem to have occurred due to random chance. It is remarkable that GK Csillag was for visual symptoms among the more susceptible cultivars, but one of the best in counting for DON.

The ANOVA of the three traits (Table 2) showed highly significant main effects. As the four-way interaction was significant in all tests, we also calculated the F-value for the AxBxCxD interaction, but basically the result remained very similar to what we found for F-values against the "Within." Beyond this, for us especially, the two-way interactions were important, which contained the main effect Genotype (A) and the other three main effects: Inoculations, Isolates, and Year (AxB, AxC, AxD). When the main effect is dominant above the two-way interaction, it means that the genotype performance is stable under the different inoculation times. Thus, in different inoculation times, even we found a significant influence, but with a modifying effect only and the probability of a very variable performance at these conditions was very low. When the main effect does not differ significantly from the interaction, it means that the stability of the main effect is poor, and we may count it with differing responses. It is also important whether the traits responded similarly or differently. There were examples for both. The Genotype (A) main effect was dominant over the AxB interaction (MS = 24.94); its level is shown by asterisks. This dominance was very strong at FHB, medium for FDK, and lowest for DON, so the traits behaved differently, even the level of significance was *p* = 0.001. For the inoculation time (B), this was opposite to AxB; for FHB data this was just above the limit of significance (as between inoculations the difference was small), but it was very high for FDK and DON, where the last inoculations gave only small DON content compared to the first three inoculation dates. The Genotype (A) main effect to AxC interaction was highly significant and the tendency corresponded to the fact we found with the Genotype main effect tested against the AxB interaction. The Isolate (C) effect was similarly dominant against AxC interaction for all traits, indicating that the main factor was the Isolate effect, being rather stable at different traits. Regarding the Inoculation date (B) interaction against the BxC interaction, we did not see significant differences except FDK; for FHB and DON the result was neutral. However, the Isolate (C) effect was highly significant for all traits against BxC interaction, but the smallest number was found for DON, and even it was highly significant. The Genotype effect against AxD was significant at *p* = 0.05 for FHB and FDK, but no difference with DON, indicating no special interaction between Genotype and Year. The Year effect was highly significant for all traits against AxD, showing a strong dominance of the year over the interaction, i.e., the stability of the Genotype performance in different years.

The degrees of freedom for the interactions were the same as those listed in the main ANOVA (Table 2). We counted the means for two years for all cultivars and isolates separately for each trait and then calculated the Pearson correlation coefficients for all isolates between traits (Table 6). All correlations were significant for isolate 3. None were significant for isolate 4. Two were significant for isolate 1 and only one for isolate 2. Furthermore, all were highly significant for the mean reactions. One general conclusion is certain, namely, the different isolates do not respond the same way. Therefore, the use of a single isolate can be risky. Here we should remark that any of the four isolates can be one in a paper where only one inoculum was used. For a publication where only one was available for phenotyping, a large variation might occur, and the genetic meaning of the data is instable. This is valid for mapping work, but also for resistance evaluation of the genotypes in breeding or cultivar registration.

## 4. Discussion

### 4.1. Susceptibility Window

Despite the fact that the susceptibility window or the flowering indicated susceptibility is well known in the *Fusarium* literature, Selby and Manns [31], as cited by Stack [32] and Atanasoff [5], mentioned it as a susceptible stage, gave no details were given. Several papers analyzed the whole period between flowering and early ripening, e.g., del Ponte et al. [11], with six inoculations from flowering to late stages of kernel development, but it was not detailed for the early SW period. The largest toxin DON contamination was found at the watery milk stage with 98 mg/kg, the smallest was only 1.2 mg/kg at the end of the vegetation period. For the disease index, here the 16-day long testing period was rather stable, with nearly no significant difference between the inoculation day’s means. This means that on this basis the identification of the SW was not possible. For FDK and DON, a sharp decrease was found in all genotypes between the 8th and 11th day. This would mean that late palea and lemma inoculations were successful, but its spreading to the developing grains was inhibited by an unknown effect. We should realize that the SW could be identified only by the FDK and DON data. The stability for DON was the most useful in the first eight days. As DON is the ultimate trait most needed, this is good news for the artificial inoculation. This means that during the first eight days we can pool the genotypes flowering in the first six days, for another we can pool those that flower between the 7th and 12th days, and the rest can be inoculated when the latest flower. As inoculations should not be made on every second or third day, they can be pooled to one day. The background weather noise in the data will be smaller and the data will be more comparable. In the last years we used two or maximum three inoculation dates [24] and a scientific control test series was needed, how this modification influences the exactness of the experi9ment5ation. The conclusion is that two inoculations were suitable when the season is warmer; in cooler weather three inoculations were needed. It might happen that ecological conditions for the two inoculation dates were similar thereafter. When the variation in the first and second group have similar means, similar minimum and maximum values, then the whole data basis can be pooled. If not, for a resistance test where earlier, middle flowering or late controls are included, this makes no problem in selection, but creates a problem when comparing the resistance of the different flowering groups. This pooled inoculation time can also help in fungicide tests, where cultivars having a difference of 4–5 days in flowering can be inoculated on the same day, when similar flowering-time genotypes could not be applied.

### 4.2. Resistance Expression

Resistance differences were large and mostly similar across the six inoculation dates. As we had in regular tests only one inoculation time, we cannot see variation in responses, but basic differences between years and isolates were not found. The data showed that differences were the largest for the early inoculations, later the differences become smaller. Resistance is measured so the behaviors of the cultivars for the visual reaction (D) can be evaluated. This corresponds to the general assumption that the resistance should be evaluated by the disease symptoms of the heads. A great number of papers using this approach prove the strength of this idea clearly. It has long been clear that the fine regulation of FDK and DON might follow different patterns. First, we recognized that they are not absolutely interdependent [3,23], indicating that even these three traits may correlate well, a part of the genotypes react differently and this allowed us to describe specific resistance traits to FDK and DON. Mesterhazy et al. [4] showed that about 80% of the decrease in DON is a consequence of the higher resistance, but other resistance-independent genetic regulation(s) exists. For this reason, it is not surprising that the same FHB values and different FDK and DON values can be seen (Figure 3). As the data show the means of six inoculation times, the supporting power of the data is larger than it would be for only one single isolate. As in visual symptoms, no clear tendency exists, and we can only state that differences in resistance exist, corresponding with earlier data. However, for FDK, five genotypes had stable performance during the first three inoculations, but GK Garaboly and GK Csillag produced larger differences. We do not think this happened by random chance. From the fourth inoculation the three more resistant genotypes did not show large variation, but afterwards a continuous decrease was seen. The DON data again showed a differing picture. Between the first four inoculations we had significant correlations, but the correlations for DON between the first four and the last two inoculations did not show a significant relationship. For example, GK Csillag had 10 mg/kg toxin content for the first inoculation, but afterwards had only 0.64 mg/kg for the sixth inoculation. This is nearly the same value (0.57 mg/kg) that was found for the most resistant genotype F569/Kő, which had only 2.98 mg/kg DON for the first inoculation. Such behavior of GK Csillag was hypothesized earlier, but the first scientific proof was produced in this paper. Such tests may contribute to better descriptions of the resistance behavior of highly important cultivars for FDK and DON, even in a regular screening it is not enough to present data to understand and explain the "irrational" behavior of the given cultivar. Such a trait may have great economic importance; it is not only a scientific result. Even GK Csillag and F569/.Kö had nearly the same DON contamination on the 16th day, the F569/.Kő resists much better in the early, highly susceptible phase. Csillag and F569/.Kő can be protected effectively by fungicides with lower DON content than the 1.25 mg/kg EU limit. With higher susceptibility without the extra ability of GK Csillag to reduce DON, the effect of fungicide efficacy can be problematic.

We have recognized that very young, strongly differentiating tissues are more susceptible, as seen in young roots during germination [26,33], and this paper together with many others, starting from Atanasoff [5], indicate a high susceptibility in the flowering time. Older tissues are less sensitive as demonstrated by del Ponte et al. [11], Cowger and Arrellano [12], and Siou et al. [13]. This study showed also that the very young head tissues are more severely infected than the later inoculated ones. However, the lemma and palea keep their susceptibility in the first 16 days, so their infection was continuously at a high level without expressed change in any direction. It seemed that the grains differed from this, since after a week (here eight days) a significant decrease in the infection severity was found by a mean 50% reduction. Thus, the data support our earlier finding [23] that FDK is regulated not only by the visual symptoms, but also by other mechanisms not known in detail. The toxin contamination decreased most intensively, seeming that aging was the most important role here. However, beside the physiological and epidemiological responses, genetic effects can also be discovered. To continue these investigations, breeders can have more profound information about their varieties, and when growers know about is, they can request such cultivars.

It seems that resistance expression is much more complex than generally supposed. Without mapping FDK and DON, a reliable picture of the genetic background of the genotypes is not possible. The differences might be explained not only by sometimes poor methodology or environment, but genetic effects also play a significant role.

### 4.3. Advantage of Using More Isolates

Most papers use a single inoculum for resistance tests, most often mixtures from different isolates. As the *Fusarium* head blight pathogens do not have specialized races [20,23,34,35] such as yellow rust, this seems reasonable. However, the experimental data gave variable results for the different isolates [34]. It seems that there is a general assumption that the aggressiveness of an isolate is stable. However, this cannot be tested when we have only one single inoculum. It is known that between years we have large differences for the same isolate. When we have more parallel isolates, the interest is in the ranking of the isolates, as this shows really the changing aggressiveness [3,4,21]. The source of the differences is the differing aggressiveness of the individual inocula, either in the case of pure isolates or mixtures. The correlations and isolate specific means showed this trend clearly (Table 6). The data also showed remarkably that every isolate has a different pattern to cause head symptoms, FDK or DON. As we do not have stability in aggressiveness across years and isolates, we do not have the same infection patterns for the different isolates and as a consequence we receive highly variable correlations between traits in different isolates. The risk is high, working with one inoculum to receive less reliable results. For research and phenotyping, we need exact data as far as possible. This is the reason that we use four independent inocula in such tests. Ecologically, the results are comparable as they were measured under the same ecological conditions, so an environment/epidemic severity interaction within a specific year is not the case. However, 2–4 years of study is necessary because between years and other sources of variance we have mostly significant interactions. There is an additional practical necessity. When projects or contract work are done, the probability is high to get low infection pressure and the test can be less successful than it should be. For this, a high infection pressure is very important. The chance is very low that not all isolates will give the accepted results in a year. When one or two fail, the test can be yet successful. With a single isolate, however, the year should be repeated, if it is possible at all.

The possible exact phenotyping is the key factor in breeding, genetic studies, and all branches of research where interactions between plant and any other influencing agents are studied. Less reliable resistance data may lead to under- or overestimation of resistance, may cause QTL artifacts in research, etc. To improve the quality of the work, the new approaches may help further both theoretical genetic research and practical breeding.

### 4.4. Resistance Testing Aspects

The pooling of inoculation dates makes it possible for treating a larger population in an inoculation and genotype differences can be identified better. The aggressiveness of the inocula for large scale tests should be checked before use. In this way, the use of inocula with standardized conidium concentration with low or no aggressiveness can be avoided. We need less inoculation times; the results will be more comparable. As resistance expression differs for different isolates, the use of more isolates gives more reliable data on resistance level and its reliability, therefore representing a higher phenotyping quality where it is needed. All published reports stress the need for low DON contamination. As DON contamination does not correlate well in about 20% of the genotypes where DON overproduction and relative DON reduction [4] occurs, its testing should be included. Therefore, serial testing for DON in an advanced stadium of breeding is important to discard toxin overproducers and highly susceptible genotypes during breeding and variety registration tests [24].

## Figures and Tables

**Figure 1 microorganisms-08-00627-f001:**
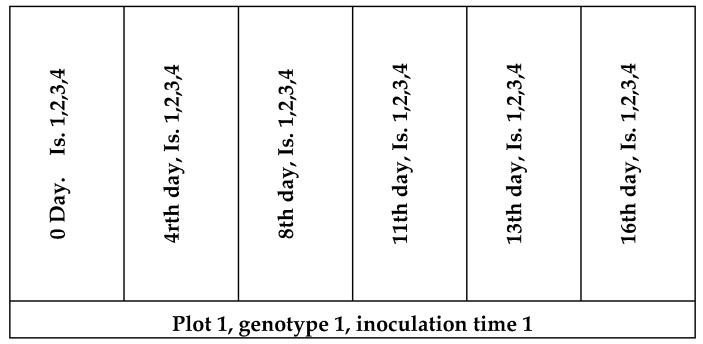
Map of a plot in the experiment. Isolates: 1, 2, 3, 4, inoculated bunches about 50 cm from each other.

**Figure 2 microorganisms-08-00627-f002:**
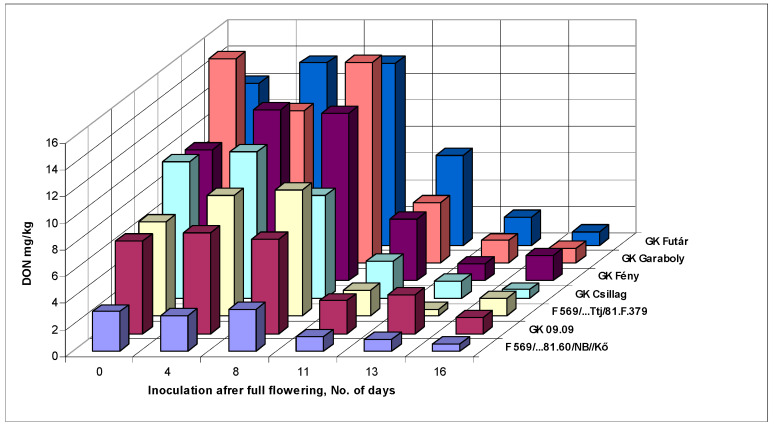
Susceptibility window against FHB in wheat, DON contamination of the wheat genotypes at different inoculation dates, Szeged, 2013 and 2014. LSD 5% genotypes: 1.12; inoculation days: 1.04.

**Figure 3 microorganisms-08-00627-f003:**
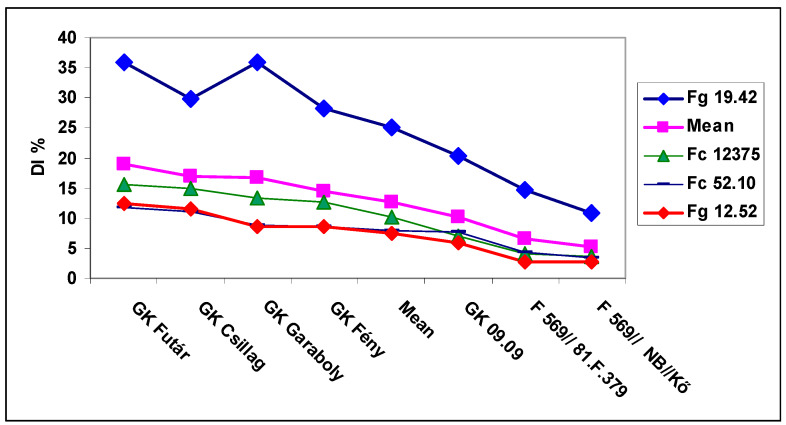
Susceptibility window in wheat, disease index (DI) % general means for the four isolates in the seven genotypes differing in resistance, 2013 and 2014, Szeged. LSD 5% between any data: 2.01.

**Figure 4 microorganisms-08-00627-f004:**
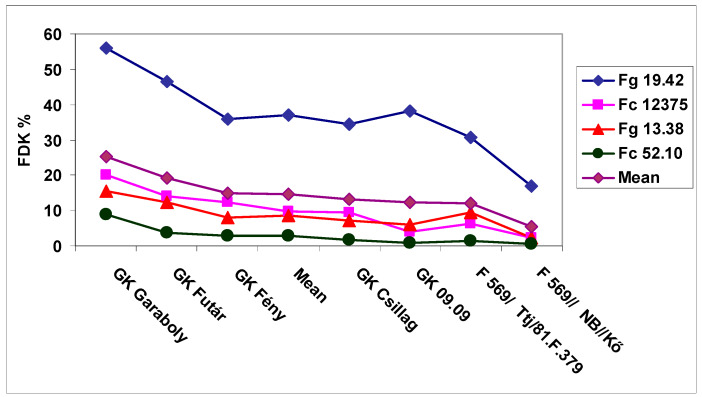
Susceptibility window in wheat, FDK % for the general means of the four isolates in the seven genotypes differing in resistance, 2013 and 2014, Szeged. LSD 5% between any data: 3.81.

**Figure 5 microorganisms-08-00627-f005:**
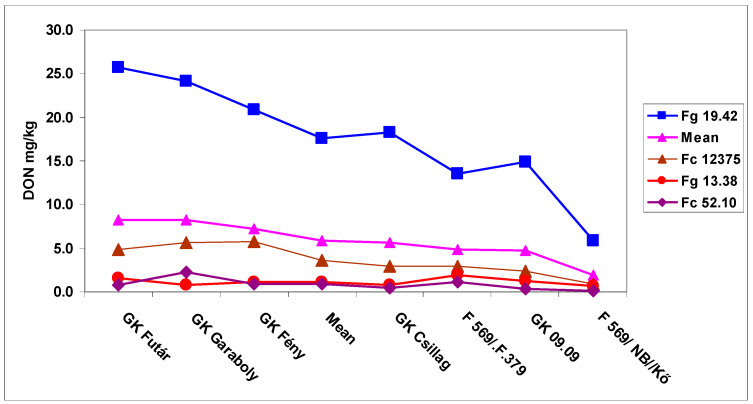
Susceptibility window in wheat, DON mg/kg values for the general means of the four isolates in the seven genotypes differing in resistance, 2013 and 2014, Szeged. LSD 5% between any data: 2.25.

**Figure 6 microorganisms-08-00627-f006:**
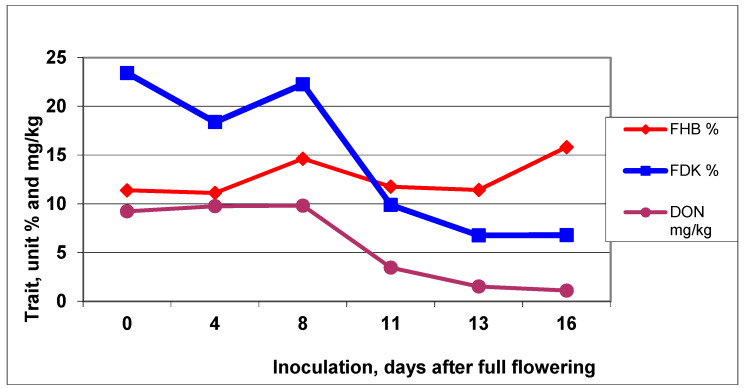
FHB disease index, FDK and DON means across genotypes at six inoculation time points during 2013 and 2014, Szeged

**Figure 7 microorganisms-08-00627-f007:**
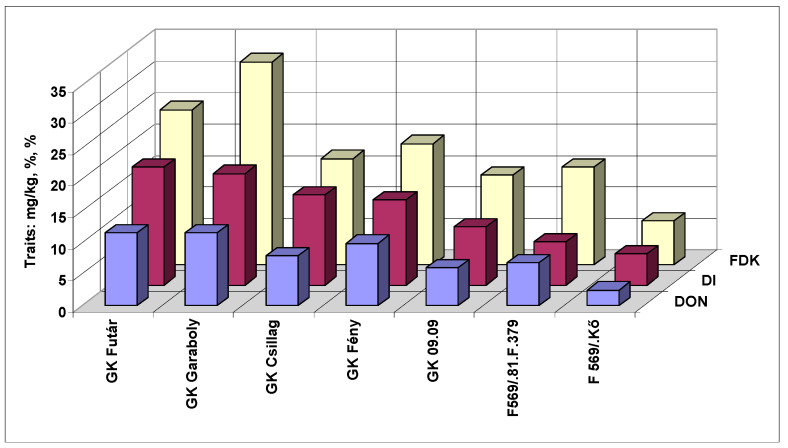
Susceptibility window against FHB in wheat showing the general pattern for genotypes and traits FHB (DI) %, FDK %, and DON mg/kg, 2013 and 2014, Szeged

**Table 1 microorganisms-08-00627-t001:** Genotypes, their abbreviations and resistance classification Szeged, 2013–2014.

Genotype	Resistance Class	Abbreviation
F 569//Ttj/RC103/3/Várkony/4/Ttj/RC103/3/81.60/NB//Kő	R	F569/Kő
F 569//Ttj/RC 103/3/Várkony/4/Ttj/81.F.379	R	F569/81/F379
GK 09.09	MS	GK 9.09
GK Fény	MR	GK Fény
GK Garaboly	S	GK Garaboly
GK Csillag	MR	GK Csillag
GK Futár	S	GK Futár

**Table 2 microorganisms-08-00627-t002:** ANOVAs for disease index, FDK, and DON contamination for the susceptibility window experiment, 2013 and 2014, Szeged.

Source of Variance	df	FHB			FDK			DON		
variance		MS	F	F _AxBxCxD_	MS	F	F _AxBxCxD_	MS	F	F _AxBxCxD_
Genotype A	6	4214.2	223.2 ***	166.3 ***	5524.5	81.07 ***	45.4 ***	738.7	31.1 ***	13.5 ***
Inoculations B	5	680.1	36.0 ***	26.8 ***	9934.2	145.7 ***	81.6 ***	3003.5	126.7 ***	55.2 ***
Isolate C	3	17680.0	936.5 ***	697.9 ***	58206.4	854.2 ***	478.6 ***	15941.6	672.4 ***	293.2 ***
Year D	1	14002.3	741.7 ***	552.7 ***	29487.3	432.7 ***	242.4 ***	12027.7	507.3 ***	221.2 ***
AxB	30	169.0	8.95 ***	6.6 ***	432.4	6.3 ***	3.5 ***	104.0	4.3 ***	1.9 **
AxC	18	400.9	21.2 ***	15.8 ***	774.9	11.3 ***	6.3 ***	363.6	15.3 ***	6.6 ***
AxD	6	740.8	39.2 ***	29.2 ***	1258.9	18.4 ***	10.3 ***	429.4	18.1 ***	7.8 ***
BxC	15	660.6	34.9 ***	26.0 ***	1250.9	18.3 ***	10.2 ***	1194.9	50.4 ***	21.9 ***
BxD	5	604.3	32.0 ***	23.8 ***	3329.4	48.8 ***	27.3 ***	1090.1	45.9 ***	20.0 ***
CxD	3	6059.2	320.9 ***	239.2 ***	20472.8	300.4 ***	168.3 ***	7674.9	323.7 ***	141.1 ***
AxBxC	90	45.9	2.4 ***	1.8 ***	109.6	1.6 ***	0.90ns	63.3	2.6 ***	1.2ns
AxBxD	30	114.6	6.06 ***	4.5 ***	337.0	4.9 ***	2.7 ***	79.4	3.3 ***	1.5 ns
AxCxD	18	80.0	4.2 ***	3.1 ***	249.7	3.6 ***	2.05 ***	269.0	11.3 ***	4.9 ***
BxCxD	15	160.4	8.4 ***	6.3 ***	902.7	13.2 ***	7.42 ***	488.1	20.5 ***	8.9 ***
AxBxCxD	90	25.3	1.34 *	1.0 ns	121.6	1.78 ***	0.99 ns	54.4	2.29 ***	1.0 ns
Within	672	18.9			68.1			23.7		
Total	1007									
Interactions			*p*			*p*			*p*	
A/AB	*df*: 6 /30	24.94	***		12.08	***		7.10	***	
B/AB	*df*: 5 /30	4.02	**		22.97	***		28.88	***	
A/AC	*df*: 6/18	10.51	***		6.74	***		2.03	ns	
C/AC	*df*: 3/18	44.10	***		75.11	***		43.84	***	
B/BC	*df*: 5/15	1.03	ns		7.94	***		2.51	ns	
C/BC	*df*: 3/15	26.76	***		46.53	***		13.34	***	
A/AD	*df*: 6/6	5.69	*		4.39	*		1.720	ns	
D/AD	*df*: 1/6	18.90	***		23.42	***		28.01	***	

*** *p* = 0.001, ** *p* = 0.01, * *p* = 0.05, ns = nonsignificant.

**Table 3 microorganisms-08-00627-t003:** Susceptibility window for *Fusarium* head blight (FHB) in wheat, disease index data, as the % of infected spikelets, across four independent isolates, 2013–2014, Szeged.

Genotype	Inoculation Days after Mid-Flowering Data DI %						Mean
	0	4	8	11	13	16	
F 569/Kő	3.81	5.76	5.50	5.04	5.58	5.39	5.18
F 569/81.F.379	7.26	4.78	8.52	7.08	3.91	7.40	6.49
GK 09.09	9.43	8.71	10.62	8.30	10.19	14.35	10.27
GK Fény	10.20	12.46	17.29	14.27	14.69	18.15	14.51
GK Garaboly	19.92	16.64	23.09	11.27	11.11	17.98	16.67
GK Csillag	13.32	12.82	15.78	15.47	17.66	26.03	16.85
GK Futár	15.80	16.67	21.66	20.90	16.83	21.49	18.89
Mean	11.39	11.12	14.64	11.76	11.42	15.83	12.69
Limit of significant difference (LSD) 5% variety							0.72
LSD 5% inoculation date							0.67
**A/Corr. between cultivars**	**F 569/Kő**	**F 569/81.F.379**	**GK 09.09**	**GK Fény**	**GK Garaboly**	**GK Csillag**	**GK Futár**
F 569/81.F.379	−0.3721						
GK 09.09	0.1879	0.2855					
GK Fény	0.6138	0.3269	0.7156				
GK Garaboly	−0.2147	0.6670	0.3078	0.1273			
GK Csillag	0.2694	0.1744	0.9295 **	0.7611 *	−0.0281		
GK Futár	0.3364	0.6620	0.4655	0.8466 *	0.1810	0.5522	
**B/Corr. between Inoculations**	**0**	**4**	**8**	**11**	**13**	**16**	
4	0.9210 **						
8	0.9458 **	0.9745 ***					
11	0.6545	0.8210 *	0.7924 *				
13	0.6178	0.8072 *	0.7287	0.9025 **			
16	0.7372	0.8223 *	0.7736 *	0.8533 *	0.9592 ***		
Mean	0.8791 **	0.9626 ***	0.9392 **	0.9072 **	0.9096 **	0.9351 **	

*** *p* = 0.001, ** *p* = 0.01, * *p* = 0.05.

**Table 4 microorganisms-08-00627-t004:** Susceptibility window for FHB in wheat, FDK data as percentage across fours independent isolates, 2013–2014, Szeged.

Genotype	Inoculation Days after Mid-Flowering Data: %						Mean
**A/Original data**	**0**	**4**	**8**	**11**	**13**	**16**	
F 569/Kő	8.96	7.90	8.06	2.88	2.83	1.95	5.43
F 569/81.F.379	18.21	17.00	18.98	7.63	4.13	6.17	12.02
GK 09.09	22.02	12.02	14.00	8.83	9.08	8.04	12.33
GK Csillag	21.86	15.46	20.60	9.27	5.90	5.96	13.18
GK Fény	22.63	21.17	22.31	10.77	5.28	6.65	14.80
GK Futár	28.98	28.09	28.17	13.10	8.82	7.88	19.17
GK Garaboly	41.29	27.04	43.75	16.71	11.29	10.83	25.15
Mean	23.42	18.38	22.27	9.89	6.76	6.78	14.58
LSD 5% genotype							1.90
LSD 5% inoculation							1.76
**B/Corr. between Cultivars**	**F 569/Kő**	**F 569/81.F.379**	**GK 09.09**	**GK Csillag**	**GK Fény**	**GK Futár**	
F 569/81.F.379	0.9723 ***						
GK 09.09	0.8371 *	0.7689 *					
GK Csillag	0.9624 ***	0.9681 ***	0.8731 *				
GK Fény	0.9742 ***	0.9942 ***	0.7781 *	0.9682 ***			
GK Futár	0.9884 ***	0.9873 ***	0.7802 *	0.9595 ***	0.9945 ***		
GK Garaboly	0.9308 **	0.9471 **	0.8370 *	0.9891 ***	0.9378 **	0.9260 **	
**C/Corr. between Inoculations**	**0**	**4**	**8**	**11**	**13**		
4	0.8473 *						
8	0.9618 ***	0.8851 **					
11	0.9832 ***	0.9088 **	0.9492 **				
13	0.8960 **	0.6409	0.7512	0.8553 *			
16	0.9412 **	0.7517	0.8409 *	0.9310 **	0.9159 **		

*** *p* = 0.001, ** *p* = 0.01, * *p* = 0.05.

**Table 5 microorganisms-08-00627-t005:** Susceptibility window for FHB in wheat, DON contamination, mg/kg across four independent isolates, 2013–2014, Szeged.

Genotype	Inoculation, Days after Mid-Flowering, Data: mg/kg						Mean
**A/Original Data**	**0**	**4**	**8**	**11**	**13**	**16**	
F 569/Kő	2.98	2.68	3.17	1.08	0.92	0.57	1.90
GK 09.09	6.96	7.57	7.10	2.47	2.94	1.21	4.71
F 569/81.F.379	7.06	9.05	9.46	1.94	0.48	1.30	4.88
GK Csillag	10.27	11.01	7.71	2.80	1.28	0.64	5.62
GK Fény	9.84	12.84	12.54	4.62	1.25	1.90	7.16
GK Garaboly	15.36	11.44	15.05	4.53	1.69	1.12	8.20
GK Futár	12.15	13.71	13.66	6.73	2.11	1.01	8.23
Mean	9.23	9.76	9.81	3.45	1.52	1.11	5.81
LSD 5% genotype							1.12
LSD 5% inoculation							1.04
**B/Corr. between Cultivars**	**F569/Kő**	**GK 09.09**	**F569/81.F.379**	**GK Csillag**	**GK Fény**	**GK Garaboly**	
GK 09.09	0.9740 ***						
F 569/81.F.379	0.9590 ***	0.9516 ***					
GK Csillag	0.9315 **	0.9619 ***	0.9239 **				
GK Fény	0.9497 **	0.9478 **	0.9913 ***	0.9362 **			
GK Garaboly	0.9894 ****	0.9389 **	0.9334 **	0.9184 **	0.9285 **		
GK Futár	0.9579 ***	0.9499 **	0.9591 ***	0.9439 **	0.9829 ***	0.9488 **	
**C/Corr. between Inoculations**	**0**	**4**	**8**	**11**	**13**	**16**	
4	0.8372 *						
8	0.9035 **	0.8760 **					
11	0.7830 *	0.8575 **	0.8551 *				
13	0.2636	0.1840	0.1688	0.3784			
16	0.2300	0.4884	0.5361	0.3456	0.0534		

**** *p* = 0.0001, *** *p* = 0.001, ** *p* = 0.01, * *p* = 0.05.

**Table 6 microorganisms-08-00627-t006:** Susceptibility window against FHB in wheat, 2013 and 2014. Correlation between different traits and the corresponding trait values. For orientation, the general means of the data are given.

Isolates	Correlations			Means for Isolates		
	DI/FDK	DI/DON	FDK/DON	FHB DI%	FDK%	DON mg/kg
1 F.c. 12375/1	0.774 *	0.732	0.904 **	10.19	9.80	3.62
2 F.c. 52.10 /2	0.427	0.173	0.916 **	7.48	2.91	0.88
3 F.g. 19.42	0.863 *	0.959 ***	0.899 **	25.13	36.93	17.60
4 F.g. 13.38/4	0.499	−0.062	0.262	7.96	8.69	1.16
Mean	0.771 *	0.869 *	0.921 **	14.58	12.69	5.81

*** *p* = 0.001, ** *p* = 0.01, * *p* = 0.05; DI = disease index; F.c.: *F. culmorum*; F.g.: *F. graminearum.*

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
