# Peer review of "Methodical Considerations and Resistance Evaluation against Fusarium graminearum and F. culmorum Head Blight in Wheat. Part 3. Susceptibility Window and Resistance Expression"

_microorganisms, 2020, doi:10.3390/microorganisms8050627_

Round 1

Reviewer 1 Report

Dear Authors, 

The subject is very relevant. But manuscript has the following problems: 

  • need very series english editting
  • no clear objective
  • no meaningful scientific experimental design
  • poor in statstical analysis
  • unclear conclusion
  • very ambiogous, long and unclear sentences

Few examples: 

Page 1

Ln 10-12: need paraphrasing 

Ln 20-21: the sentence does not make sense. 

Ln 28-30: need paraphrasing 

Ln 32-33, 43-44: ambiguous sentence 

Page 3:

Ln 90-96: very har to understand 

Ln 105-112: it is a very messy paragraph which is hard to classify as objective or Method or introduction 

ettc.

Need major modification.

Author Response

Answer to Reviewer 1.

Thanks for the careful opinion. We screened the whole paper as in your table all chapters of the paper were mentioned to improve. The individual remarks were answered printed with tracks. We hope that corrections made correspond to your intentions. In the case you find definite problems, we are ready to answer them as good as possible.

Yours sincerely

Comments and Suggestions for Authors

Dear Authors, 

The subject is very relevant. But manuscript has the following problems: 

  • need very series english editting
  • I have the message from the Editors, that a proofreadin fod English will come, when the paper is so far that only this will be needed to accept it. For this reason following this survay, the paper will be edited in this respect.
  • no clear objective
  • It is corrected
  • no meaningful scientific experimental design
  • It is corrected
  • poor in statstical analysis. To this I need some help, we made a correct four-way ANOVA . I see seldom such analyses printed. When you would suggest me a better one, I would do it additionally. When needed, I could send the statistical evaluations in detail for all traits.
  • It was added, the Materials and Methods were extended.
  • unclear conclusion
  • It was made.
  • very ambiogous, long and unclear sentences
  • The long sentenced were broken to shorter ones, in some cases part omitted, some word added. I hope, it will be much clearer.
  • We have extended the Materials and Methods with Inoculatin, DON analysis

Few examples: 

Page 1

Ln 10-12: need paraphrasing  Done

Ln 20-21: the sentence does not make sense. Done

Ln 28-30: need paraphrasing Done

Ln 32-33, 43-44: ambiguous sentence Done

Page 3:

Ln 90-96: very har to understand Done

Ln 105-112: it is a very messy paragraph which is hard to classify as objective or Method or introduction  Done

ettc.

Need major modification. It was done. Thanks.

Reviewer 2 Report

The study authored by Gyorgy et al, evaluated the suitability of disease index (DI), Fusarium damaged kernel rate (FDK) and deoxynivalenol (DON) as susceptibility window. They showed that the DI was not suitable to determine length of SW. They demonstrated that FDK and DON are relatively stable traits to asses SW. In addition, they also suggested that inoculation of more isolates can improved the reliability of the testing.

The research topic is interesting. The experiments are well designed, and the data are properly analyzed. The data from the study provide information for evaluation disease and toxin resistant varieties and the time of applying fungicide. My overall comments: Many sentences are not concise and clear. I highlight some, correct a few. I suggested that the authors read through their manuscript carefully and make correction. It will be helpful if they can have a native English speaker to read through and make corrections.

See attached file.

Author Response

Comments and Suggestions for Authors
The study authored by Gyorgy et al, evaluated the suitability of disease index (DI), Fusarium damaged kernel rate (FDK) and deoxynivalenol (DON) as susceptibility window. They showed that the DI was not suitable to determine length of SW. They demonstrated that FDK and DON are relatively stable traits to asses SW. In addition, they also suggested that inoculation of more isolates can improved the reliability of the testing.
The research topic is interesting. The experiments are well designed, and the data are properly analyzed. The data from the study provide information for evaluation disease and toxin resistant varieties and the time of applying fungicide. My overall comments: Many sentences are not concise and clear. I highlight some, correct a few. I suggested that the authors read through their manuscript carefully and make correction. It will be helpful if they can have a native English speaker to read through and make corrections.
See attached file.
In the attached paper with your remarks I found a request about the concentration of inoculum. I inserted an explanation, but as this problem is not a part of the paper, it clears why we choose another way. Here I would like to give a little bit more information. The literature is clear, there is no standard conidium concentration, and they show up about 100 fold difference. From this we cannot have any information, which is optimal. When I chose seven million or 40 thousand, both are equally good and supported with good articles in highly estimated journals. Originally the conidium concentration was thought to regulate aggressiveness. For this we have sources. As in the literature to a given conidium concentration many different disease level is attached, it is clear that the use of a given conidium concentration does not have any meaning for aggressiveness, no author present any data or aim to give a reason why the given conidium concentration was used. The next problem is that most isolates of F. graminearum have low or poor conidium producing ability, e.g. in liquid media beside conidia also smaller or larger amount of mycelium is produced. As between conidia and mycelium no difference exist in disease causing capacity, we could also measure the total fungal mass of the suspension. This can be done easily by a spectrophotometer and this has the advantage that also inocula with mycelium content could to be quantified. The aggressiveness and the rate of conidia and mycelium is also a genetically regulated trait, the conidium concentration is not good for that as among high and low conidium producing isolates high differences in aggressiveness can be shown. The regulation has the ultimate task to secure high aggressiveness for the experiments.
This is the reason we test directly aggressiveness that tells us what the usefulness of the given inoculum is. Most methods that produce large quantities of infection material, are mixed, conidia and mycelium are present. To separate from 700 L inoculum the conidia, nobody will do in a breeding or testing station. Even in mungo bean agar where the conidium production is better. They want one thing, to know, how infective the given inoculum is. When this is not possible, they use spam inoculation, or like Mike Taylor in Limagrain, who collects the after screening the small and scabby grains, he mills that, gives regular water in given ratio to that and next day he sprays the plants. For mass work they can be done, but a fine research work cannot be done, where for example phenotyping of a mapping population is necessary. Of course, I accept for specific investigations the standardizing fungal mass, and its scientific value would be better that the conidium concentration is and could come nearer to the needs of the praxis needs from us. The aggressiveness test is standardized, we see also the dilution ability of the given inoculum. As we need as high aggressiveness as possible, normally we use inocula without dilution. From the dilution tests we know that a dilution 1:1 with ion changed sterile water does not influence very seriously the aggressiveness, we make some when the amount is smaller than needed, for example we need 12 liter, but we have only 9.5. In this case we will use a dilution 1:0.3 dilution. This happened with these isolates we used. And, in spite of the not standardized conidium concentration, the inocula gave across the two years, significantly differing aggressiveness data. I think, in this respect we need a lot of research.

Round 2

Reviewer 1 Report

Dear Authors,

thank you for your efforts to modify the manuscript. But still it need some corrections and clarifying unclear sentences. Examples are attached and also described below:

Objectives 2 and 4 are not clear

Lines 173-175: not clear

It will be more scientific if you can add the "LSD" values 

Still the last part of the conclusion needs clarification

In general, please make your sentences very clear to the reader. present your results in a more scientific way. Still it needs more English editing before publication.

Few examples of corrections/comments are attached.

Author Response

Comments and Suggestions for Authors

Dear Rewiewer 2,

Thank you very much for your remarks. I made the thest in several sases better, some mistakes were improved. Table 5 was transferred to its final location and the Figura 2 was cirectly copied from the Excel file, so it looks now as it should be.

Dear Authors,

thank you for your efforts to modify the manuscript. But still it need some corrections and clarifying unclear sentences. Examples are attached and also described below:

Objectives 2 and 4 are not clear. Rewritten

Lines 173-175: not clear Done

It will be more scientific if you can add the "LSD" values .

 All tables, where appropriate contain the LSD values, for the figures I added them, counted for the genoype/isolate interaction..

Still the last part of the conclusion needs clarification

I think, you are right. This is not a breeding paper, therefore I concentrated on the methodical points that are very similar you have written in your first opinion. This was a good point.

In general, please make your sentences very clear to the reader. present your results in a more scientific way. Still it needs more English editing before publication.

A language controll will be made , I hope during this problems will be solved. Thereafter you will see again the latests version, so we can thereafrer modify what should yet be done.